# Structural basis of substrate recognition and membrane association by the bacterial lysyl-phosphatidylglycerol hydrolase AcvB
Mizuki Hoshi[1], Daiki Matsumoto[1] & Yasunori Watanabe ⦿[2] ✉

Bacteria adapt to environmental stresses via membrane phospholipid remodeling; however, the underlying molecular mechanism remains largely elusive. In *Agrobacterium tumefaciens*, the lysyl-phosphatidylglycerol (Lys-PG) synthase *lpiA* and periplasmic hydrolase *acvB* genes form an operon that controls Lys-PG levels. We determined the crystal structures of mature AcvB and its C-terminal catalytic domain at 3.1 Å and 1.8 Å resolution, respectively. The catalytic domain forms a negatively charged cavity that recognizes the positively charged Lys-PG head group through multiple acidic residues, including Asp271, Asp340, and Asp370. A hydrophobic protruding loop containing Trp378 and Leu379 mediates membrane association and contributes to Lys-PG hydrolysis. Further, AcvB interacts with LpiA via its C-terminal domain, suggesting a cooperative module for Lys-PG turnover. These findings reveal the structural basis of Lys-PG hydrolysis and provide mechanistic insight into adaptive lipid modification at the bacterial membrane interface, and may guide future development of antibacterial agents against plant-pathogenic bacteria.

Biological membranes are composed of diverse phospholipids that not only provide structural integrity but also play critical roles in cellular adaptation to environmental stresses. For example, in response to environmental temperature changes, the level of unsaturated fatty acids in membrane phospholipids is modulated to maintain appropriate membrane fluidity[1,2]. Bacterial cells can adapt in response to harmful compounds by modifying their membrane phospholipids[3]. Thus, dynamic membrane phospholipid remodeling represents a fundamental mechanism by which bacteria and other organisms adapt to environmental challenges; however, the molecular basis of these adaptive lipid modifications remains largely elusive.

Bacterial membranes mainly consist of phosphatidylethanolamine (PE) and the anionic phospholipids phosphatidylglycerol (PG) and cardiolipin and therefore have a net negative surface charge[3,4]. Cationic antimicrobial peptides (CAMPs) bind to the negatively charged bacterial membrane, inducing membrane disruption and antibacterial activity[5–7]. To resist CAMPs, many bacteria reduce the net negative surface charge by adding aminoacyl groups to PG, forming aminoacyl-PG. Aminoacyl-PGs exist in various forms, of which lysyl-PG (Lys-PG) and alanyl-PG (Ala-PG) are the most common[8–10]. Many Gram-positive bacteria such as *Staphylococcus aureus* and certain Gram-negative bacteria, including the plant

pathogen *Agrobacterium tumefaciens*, contain Lys-PG in their membranes, reducing the net negative membrane charge and conferring resistance to CAMPs[11,12]. Ala-PG is found in certain Gram-positive as well as Gram-negative bacteria, such as *Enterococcus faecalis* and *Pseudomonas aeruginosa*, respectively[8,12,13].

Aminoacyl-PGs are synthesized by multiple peptide resistance factor (MprF), which was originally identified in *S. aureus*[10]. MprF is a bifunctional protein; its cytoplasmic domain functions as an aminoacyl-PG synthase that uses PG and aminoacyl-tRNA as substrates, whereas its membrane-embedded domain acts as a flippase that translocates aminoacyl-PG to the outer leaflet of the membrane[14]. In *A. tumefaciens*, Lys-PG is synthesized by the Lys-PG synthase LpiA, an MprF homolog[12,15]. In addition, *A. tumefaciens* has a soluble periplasmic Lys-PG hydrolase, *Agrobacterium* chromosomal virulence protein B (AcvB), which hydrolyzes Lys-PG into PG and lysine and is encoded in an operon along with *lpiA*[15,16]. The physiological importance of AcvB has been demonstrated by a genetic study in *A. tumefaciens*[16]; deletion of *acvB* resulted in elevated Lys-PG levels, a severe growth defect at low pH, and impaired tumor formation in host cells. Further, the elevated Lys-PG levels inhibited T-DNA transfer to host cells via the type IV secretion system, thereby impairing the virulence of *A.*

[1]Graduate School of Science and Engineering, Yamagata University, Yamagata, Japan. [2]Faculty of Science, Yamagata University, Yamagata, Japan. ✉e-mail: yasunori@sci.kj.yamagata-u.ac.jp

*tumefaciens*. These observations indicated that AcvB plays a crucial role in the physiological adaptation to acidic conditions and maintenance of appropriate Lys-PG levels to ensure membrane homeostasis and optimal virulence in *A. tumefaciens*.

Structural studies of LpiA/MprF have provided insights into the molecular mechanisms of substrate recognition and aminoacyl-PG flipping[17–19]. Recently, the crystal structure of the N-terminal D1 domain of the AcvB homolog VirJ from *Brucella abortus*, which is distinct from the catalytic domain, has been reported[20]. However, the precise molecular mechanisms involved in Lys-PG hydrolysis, substrate recognition, and membrane binding remain unclear. In this study, we determined the crystal structures of the mature form of *A. tumefaciens* AcvB (lacking the signal sequence) and its C-terminal catalytic domain at 3.1 Å and 1.8 Å resolution, respectively. The C-terminal catalytic domain of AcvB forms a negatively charged cavity surrounding the active site, which is responsible for the recognition of the positively charged head group of Lys-PG. Furthermore, we showed that the protruding loop region near the active site contributes to membrane binding and the recognition of Lys-PG acyl chains. These findings provide fundamental insights into the molecular mechanism of Lys-PG hydrolysis and establish a structural framework for understanding adaptive lipid modifications at the bacterial membrane interface.

## Table 1 | Data collection and refinement statistics

| | AcvB$^{\Delta N24}$ [PDB ID 9XHM] | AcvB$^C$ [PDB ID 9XHN] |
|---|---|---|
| *Data collection* | | |
| Space group | P1 | P1 |
| *Cell dimensions* | | |
| *a, b, c* (Å) | 66.57, 76.27, 77.22 | 47.19, 64.15, 68.82 |
| *α, β, γ* (°) | 113.86, 98.35, 106.78 | 85.83, 81.44, 87.70 |
| Resolution (Å) | 50.0–3.13 (3.32–3.13) | 50.0–1.83 (1.94–1.83) |
| $R_{sym}$ | 0.126 (0.994) | 0.076 (0.928) |
| $R_{meas}$ | 0.156 (1.251) | 0.094 (1.142) |
| *I/σI* | 6.5 (1.0) | 9.2 (1.2) |
| Completeness (%) | 97.2 (95.4) | 96.1 (94.9) |
| Redundancy | 2.7 (2.4) | 2.8 (2.8) |
| $CC_{1/2}$ | 0.991 (0.426) | 0.996 (0.604) |
| *Refinement* | | |
| Resolution (Å) | 40.6–3.13 | 46.6–1.83 |
| No. of reflections | 22,037 | 68,328 |
| $R_{work}/R_{free}$ | 0.225/0.280 | 0.197/0.222 |
| *No. of atoms* | | |
| Protein | 6187 | 6205 |
| Ligand/ion | 8 | 28 |
| Water | – | 385 |
| *B-factors* | | |
| Protein | 73.77 | 35.02 |
| Ligand/ion | 73.12 | 42.48 |
| Water | – | 41.73 |
| *R.m.s. deviations* | | |
| Bond lengths (Å) | 0.002 | 0.007 |
| Bond angles (°) | 0.5 | 0.9 |

A single crystal was used for each structure determination. Values in parentheses are for highest-resolution shell.

## Results

### Crystal structure of the mature form of AcvB

To obtain structural insights into the mechanism of Lys-PG hydrolysis by AcvB, we attempted to determine the crystal structure of AcvB. AcvB (456 amino acids) contains an N-terminal signal sequence for periplasmic localization, with a predicted cleavage site after residue 24 according to SignalP 6.0[21]. We thus expressed and purified AcvB(25–456), corresponding to the predicted mature periplasmic form and hereafter referred to as AcvB$^{\Delta N24}$, using the *E. coli* expression system. The recombinant protein retained Lys-PG hydrolase activity (Supplementary Fig. 1). Ser336 has been proposed to be a conserved catalytic residue in AcvB homologs[16,22]. We substituted this residue with alanine to generate the AcvB$^{\Delta N24}$(S336A) variant and assessed its Lys-PG hydrolase activity. The S336A substitution abolished the enzymatic activity, confirming the proposed role of Ser336 (Supplementary Fig. 1). Therefore, we used AcvB$^{\Delta N24}$ for crystallization and successfully obtained well-diffracting crystals. The crystal structure of AcvB$^{\Delta N24}$ was determined via the molecular replacement method, using the structure predicted by ColabFold as the search model[23,24], and subsequently refined to 3.1 Å resolution (Table 1). The asymmetric unit of the crystal contained two AcvB$^{\Delta N24}$ molecules with a similar conformation, with a root mean square deviation (r.m.s.d.) of 0.38 Å over 386 Cα atoms. In the crystal structure, AcvB consisted of two domains: the N-terminal (residues 29–248) domain and the C-terminal catalytic (residues 249–456) domain. Both adopted α/β hydrolase folds with a similar architecture (Fig. 1A and Supplementary Fig. 2A). The N-terminal domain formed a six-stranded parallel β-sheet (β1–β6) surrounded by six α-helices (α1–α6), and the C-terminal domain formed a seven-stranded parallel β-sheet (β7–β13) surrounded by eight α-helices (α7–α14). In addition, the N-terminal domain of AcvB adopted a fold similar to the N-terminal D1 domain of the AcvB homolog VirJ from *B. abortus*[20], with an r.m.s.d. value of 2.32 Å over 145 Cα atoms (Supplementary Fig. 3). Although the N-terminal and C-terminal domains of AcvB were highly similar to the AlphaFold3-predicted domain structures, with r.m.s.d. values of 0.39 Å over 174 Cα atoms and 0.37 Å over 173 Cα atoms, respectively, the overall domain orientation differed from the prediction (Supplementary Fig. 4). Ser336, a catalytic residue, was located between β10 and α10 of the C-terminal domain. In addition, the C-terminal domain harbored a protruding loop region between β11 and α12, which is located near Ser336, suggesting that its proximity to the catalytic residue may contribute to substrate recognition. Electrostatic surface analysis revealed that the region surrounding Ser336 formed a negatively charged cavity, which could accommodate the positively charged head group of Lys-PG (Fig. 1B).

### Crystal structure of the C-terminal domain of AcvB

The C-terminal domain of AcvB has been shown to be necessary and sufficient for the Lys-PG hydrolase activity[16]. To verify the Lys-PG hydrolase activity of each AcvB domain, we prepared the N-terminal (residues 25–248; AcvB$^N$) and C-terminal (residues 249–456; AcvB$^C$) domains. Consistent with the previous study, AcvB$^C$ exhibited detectable Lys-PG hydrolase activity, whereas AcvB$^N$ did not (Supplementary Fig. 5). Notably, AcvB$^C$ displayed slightly higher hydrolase activity than AcvB$^{\Delta N24}$, raising the possibility that the N-terminal domain might influence the catalytic activity. To gain a more detailed understanding of its catalytic mechanism, we next attempted to determine the high-resolution crystal structure of AcvB$^C$. AcvB$^C$ was expressed and purified using the *E. coli* expression system, and well-diffracting crystals were successfully obtained. Its crystal structure, determined at 1.8 Å resolution, closely resembled the corresponding region in AcvB$^{\Delta N24}$, with an r.m.s.d. of 0.34 Å over 178 Cα atoms (Fig. 2A). His433 is reportedly a catalytic residue, along with Ser336[16]. In the crystal structure, His433 was positioned in close proximity to Ser336, suggesting that Ser336 and His433 likely function as a catalytic dyad (Fig. 2B and Supplementary Fig. 2B). Notably, six acidic residues (Asp271, Asp340, Glu368, Asp370, Glu410, and Glu413) were also located near Ser336, suggesting a role in recognizing the positively charged head group of Lys-PG (Fig. 2B). To investigate their roles in catalytic activity, we substituted these residues with

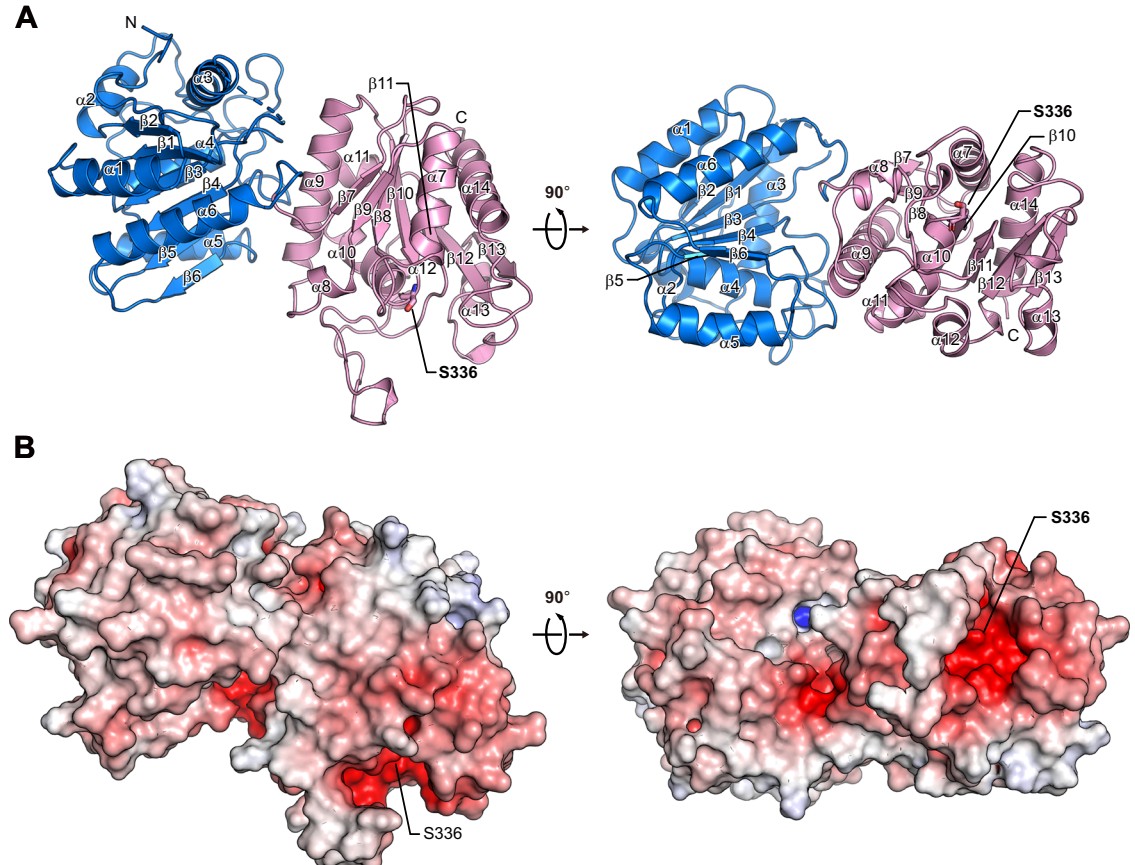

**Fig. 1 | Structure of AcvB$^{\Delta N24}$. A** Ribbon diagram of AcvB$^{\Delta N24}$ structure. The N-terminal and C-terminal domains are colored blue and pink, respectively. Secondary structural elements are labeled. The catalytic residue Ser336 is shown in stick representation. **B** Electrostatic surface potential of AcvB$^{\Delta N24}$ calculated using PyMOL. Positive and negative potentials are shown in blue and red, respectively.

alanine (D271A, D340A, E368A, D370A, E410A, and E413A). The Lys-PG hydrolase activity of the AcvB variants was assessed by co-expressing each variant with LpiA, the Lys-PG synthase of *A. tumefaciens*, in *E. coli* cells. Protein expression was confirmed using immunoblotting (Fig. 2C). Lipid extracts from *E. coli* cells co-expressing AcvB and LpiA were analyzed using thin-layer chromatography (TLC) followed by ninhydrin staining of the TLC plate to assess the levels of Lys-PG (Fig. 2D). Under the ninhydrin staining conditions, PE was predominantly detected from *E. coli* cell membranes harboring empty vectors, whereas both PE and Lys-PG were detected from those of LpiA-expressing cells. Co-expression of wild-type AcvB with LpiA in *E. coli* cells resulted in a marked decrease in the amount of Lys-PG compared with that in cells expressing LpiA alone, indicating that wild-type AcvB catalyzed Lys-PG hydrolysis. Consistent with its essential role in catalysis, the S336A substitution abolished the hydrolase activity of AcvB, resulting in the accumulation of Lys-PG in the membranes (Fig. 2D, E). Similarly, the D271A substitution caused strong Lys-PG accumulation, whereas D340A and D370A resulted in moderate accumulation. By contrast, the E368A, E410A, and E413 A variants did not accumulate Lys-PG; their Lys-PG levels were comparable to those in cells expressing wild-type AcvB. These results indicated that Asp271, Asp340, and Asp370 are critical for Lys-PG hydrolysis, whereas Glu368, Glu410, and Glu413 are not essential for the catalytic activity.

### Structural basis of substrate Lys-PG recognition

To explore the structural basis of substrate recognition, we attempted to co-crystallize AcvB with Lys-PG. Despite numerous crystallization trials, crystals of the complex could not be obtained. Therefore, we instead carried out docking simulations using AutoDock Vina[25] to model the binding of a lysyl-glycerol head-group fragment, representing the polar head group of Lys-PG, to AcvB$^C$. The lysyl-glycerol fragment was docked into a cavity corresponding to the putative substrate-binding pocket of AcvB (Fig. 3A). In the docking model, the carbonyl carbon of the ester group was positioned in close proximity to the catalytic residue S336 (4.1 Å), consistent with a plausible model for substrate binding and catalysis. The lysine moiety of the fragment was positioned near Asp271, Ser336, Phe337, Asp340, Glu368, Asp370, and Glu384. The side chains of Asp271 and Ser336 were within a distance that allowed hydrogen bonding with the α-amino group of lysine, whereas the side chains of Asp370 and Glu384 were able to form hydrogen bonds with its ε-amino group. The glycerol moiety of the fragment lied adjacent to Leu365, Ser366, Glu410, Glu413, Glu414, and His433, with the side chain of Glu410 capable of forming hydrogen bonds with the hydroxyl groups of glycerol. Given that substitution of Asp271, Asp340, and Asp370 reduced Lys-PG hydrolysis (Fig. 2D), the docking model was consistent with the notion that recognition of the lysine moiety by these acidic residues is critical for the catalytic activity of AcvB.

To assess the specific contribution of the acidic residues to lysine recognition, we examined the hydrolysis of lysine ethyl ester (Lys-EE), a simplified substrate that mimics the lysine head group without a PG moiety. Lys-EE was expected to be hydrolyzed by AcvB, yielding lysine and ethanol (Fig. 3B). N-terminally His$_6$-tagged AcvB$^{\Delta N24}$ variants (S336A, D271A, D340A, and E410A) were prepared and used in Lys-EE hydrolysis assays (Fig. 3C). Lys-EE was incubated with the variants, and the reaction products were analyzed using TLC. Wild-type AcvB$^{\Delta N24}$ efficiently generated lysine when compared with the S336A variant, demonstrating that the hydrolysis of Lys-EE was catalyzed by AcvB$^{\Delta N24}$ (Fig. 3D, E). Both the D271A and D340A substitutions reduced Lys-EE hydrolase activity, whereas the E410A

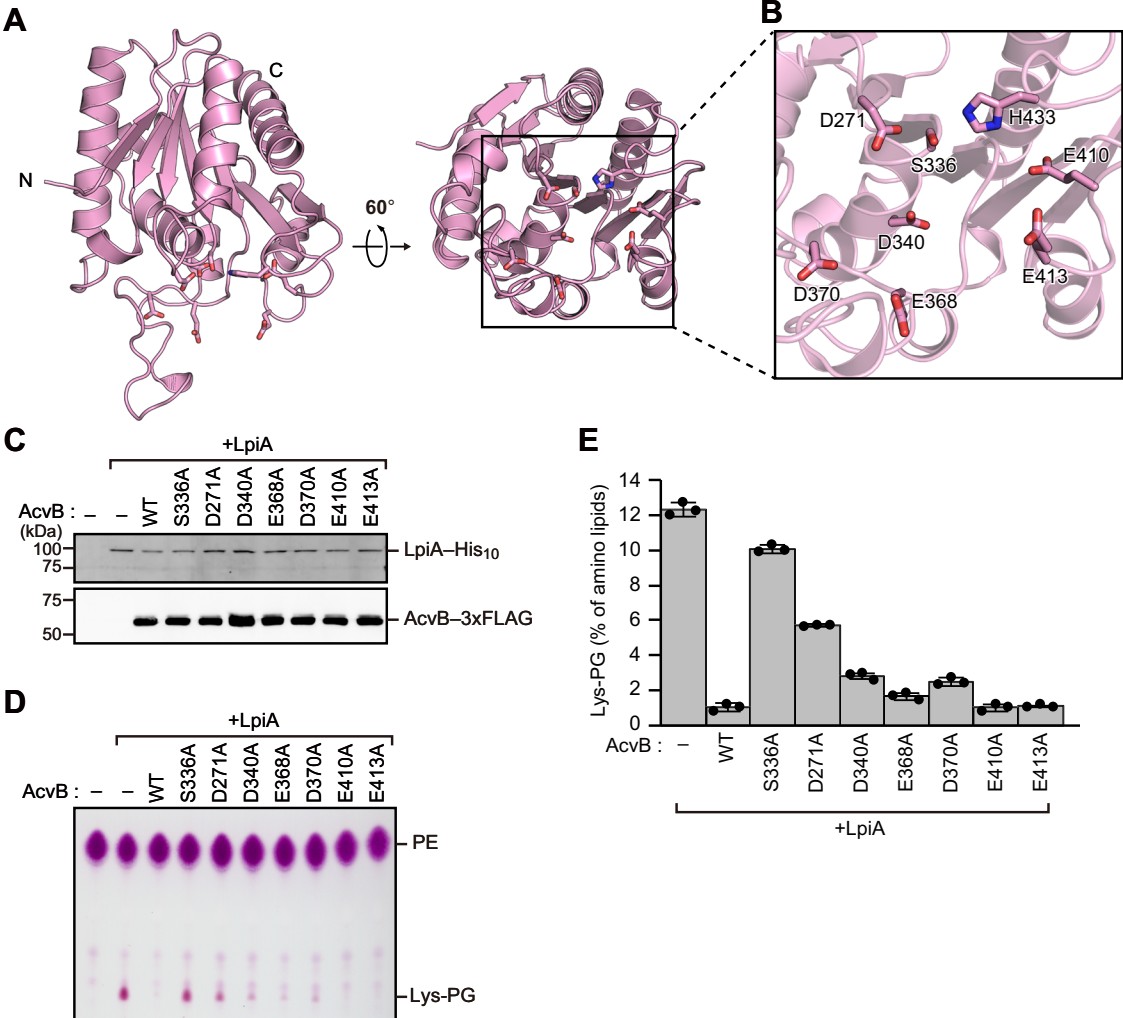

**Fig. 2 | Structure of AcvB^C. A** Ribbon diagram of AcvB^C. **B** Magnified view of the region around the active site. Asp271, Ser336, Asp340, Glu368, Asp370, Glu410, Glu413, and His433 are shown in stick representation. **C** Western blot showing relative expression levels of C-terminally His10-tagged LpiA and C-terminally 3xFLAG-tagged AcvB variants in *E. coli* cells. **D** Total phospholipids were extracted from *E. coli* cells expressing C-terminally His10-tagged LpiA and C-terminally 3xFLAG-tagged AcvB variants, separated by TLC, and visualized by ninhydrin staining. **E** Quantification of Lys-PG based on the data in (**D**), using the ImageJ software. Values represent mean ± SD from three independent experiments (*n* = 3).

variant retained activity comparable to that of the wild-type. These results supported that Asp271 and Asp340 play a critical role in lysine recognition during Lys-PG hydrolysis.

### A protruding loop region contributes to membrane association

AcvB has been described as a soluble periplasmic protein[16]; however, its substrate Lys-PG is a membrane phospholipid, implying that AcvB likely associates with the membrane during substrate recognition. In the crystal structure, one AcvB^ΔN24 molecule interacted with the N-terminal domain of a neighboring AcvB^ΔN24 molecule via a protruding loop in the C-terminal domain (Fig. 4A). Trp378 and Leu379 in the protruding loop were buried in the hydrophobic pocket of the N-terminal domain formed by hydrophobic residues such as Leu136, His206, and Leu210 (Fig. 4B). Although analysis using the PISA server[26] indicated that this interaction was a crystal packing artifact, the protruding loop was in contact with a hydrophobic region of a neighboring AcvB^ΔN24 molecule in the crystal, suggesting that it may bind to hydrophobic regions such as the membrane. To investigate how AcvB^ΔN24 interacts with the membrane, we predicted its membrane-binding orientation using the Positioning of Proteins in Membrane (PPM) server[27]. Consistent with our hypothesis, the protruding loop, particularly Trp378 and Leu379, was predicted to mediate membrane association (Fig. 4C). Taken together, these findings supported the notion that membrane

association mediated by Trp378 and Leu379 provides a plausible mechanism for Lys-PG recognition, as the loop region is positioned near the catalytic residue Ser336.

### The protruding loop region is required for membrane association and Lys-PG hydrolysis

To investigate the role of the protruding loop in catalytic activity, we constructed an AcvB variant lacking the loop region (residues 373–382; ΔLoop) (Fig. 5A). The Lys-PG hydrolase activity of the ΔLoop variant was assessed by co-expressing it with LpiA in *E. coli* cells (Fig. 5B). Similar to S336A substitution, deletion of the loop region caused Lys-PG accumulation, indicating that the loop region is important for the Lys-PG hydrolase activity (Fig. 5B, C). Trp378 and Leu379 in the protruding loop region were likely to be involved in membrane binding (Fig. 4C). To assess the role of these residues in catalytic activity, we generated two double variants in which both residues were substituted with either hydrophilic asparagine (W378N/L379N) or hydrophobic phenylalanine (W378F/L379F). The W378N/L379N variant showed Lys-PG accumulation, whereas the W378F/L379F variant did not, similar to wild-type AcvB (Fig. 5B, C). These results indicated that the hydrophobic property of Trp378 and Leu379 plays a critical role in Lys-PG hydrolase activity.

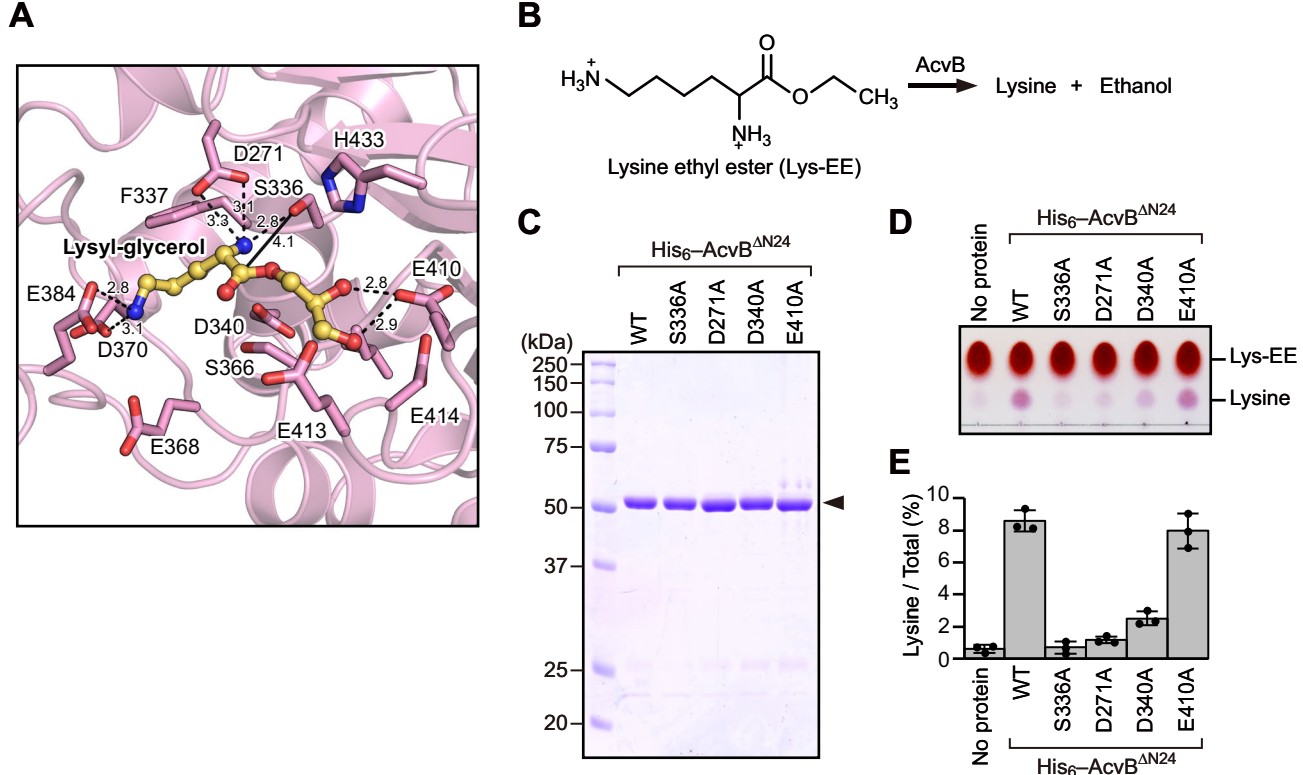

**Fig. 3 | Residues responsible for Lys-PG head-group recognition. A** Magnified view of the area around the lysyl–glycerol molecule in the docking model of AcvB[C] with a docked lysyl–glycerol. The lysyl-glycerol molecule is shown as a ball-and-stick model in yellow. Residues located near the lysyl–glycerol are shown in stick representation. The distance between the carbonyl carbon of the ester group of the lysyl–glycerol and the hydroxyl oxygen of Ser336 is indicated by a solid line. Dashed lines indicate possible hydrogen bonds. Numbers indicate distances (Å). **B** Schematic representation of the hydrolysis reaction of Lys-EE catalyzed by AcvB. AcvB hydrolyzes Lys-EE into lysine and ethanol. **C** Purified AcvB[ΔN24] variants were analyzed using SDS-PAGE followed by CBB staining. **D** Lys-EE and purified AcvB[ΔN24] variants were incubated at 37 °C for 30 min. Proteins were separated using TLC and visualized using ninhydrin staining. **E** Quantification of lysine based on the data in (**D**), using ImageJ. Values represent mean ± SD from three independent experiments ($n = 3$).

We next examined whether Trp378 and Leu379 are required for substrate recognition or membrane association during Lys-PG hydrolysis. To this end, we first tested the membrane binding of AcvB variants. N-terminally His_6-tagged AcvB[ΔN24] W378N/L379N and W378F/L379F variants were expressed in *E. coli* cells, and soluble and membrane fractions were isolated and analyzed using immunoblotting (Fig. 5D). Wild-type AcvB[ΔN24] and the W378F/L379F variant showed comparable levels of membrane association, whereas the W378N/L379N variant was present at only ~5% in the membrane fraction (Fig. 5E). Consistent with the predicted membrane-binding orientation (Fig. 4C), these findings demonstrated that the hydrophobic nature of Trp378 and Leu379 is crucial for membrane association.

To specifically assess the contributions of Trp378 and Leu379 to substrate recognition, the W378N/L379N and W378F/L379F variants were subjected to Lys-EE hydrolysis assays (Fig. 5F). Interestingly, both variants generated lysine from Lys-EE at levels comparable to that of wild-type AcvB[ΔN24], suggesting that Trp378 and Leu379 are not essential for Lys-PG head-group recognition (Fig. 5G, H). Collectively, these results indicated that Trp378 and Leu379 in the protruding loop are dispensable for recognition of the polar head group of Lys-PG but are essential for membrane association through their hydrophobic side chains.

### AcvB interacts with LpiA via the C-terminal domain

In various Gram-negative bacteria, such as *A. tumefaciens* C58, *lpiA* and *acvB* are organized in an operon[15]. Given this genomic arrangement, we hypothesized that AcvB and LpiA may functionally cooperate through direct interaction. To test this hypothesis, we examined whether AcvB physically associates with LpiA. N-terminal glutathione S-transferase

(GST)-fused AcvB[ΔN24] and C-terminally His_10-tagged LpiA were purified and subjected to an in vitro GST pull-down assay, which revealed that AcvB[ΔN24] interacted with LpiA (Fig. 6A). We next investigated whether the N-terminal or C-terminal domain of AcvB is responsible for this interaction. GST pull-down assays revealed that AcvB[C] also associated with LpiA, although this interaction was weaker than that observed for AcvB[ΔN24] (Fig. 6A, B). In contrast, AcvB[N] exhibited only a weak association with LpiA compared with both AcvB[ΔN24] and AcvB[C] (Fig. 6A, B). In a reciprocal Ni–NTA pull-down assay, His_10-tagged LpiA immobilized on Ni-NTA resin pulled down AcvB[ΔN24] and AcvB[C] but not AcvB[N] (Supplementary Fig. 6). These results indicated that the C-terminal domain predominantly mediates the interaction with LpiA, while the contribution of the N-terminal domain might be weak and assay-dependent.

To further examine whether the AcvB membrane association depends on the LpiA catalytic activity, we co-expressed a catalytically inactive AcvB S336A variant with either wild-type LpiA or the catalytically inactive LpiA D750N variant[17] (Supplementary Fig. 7A, B). We used the S336A substitution to prevent changes in Lys-PG levels resulting from the AcvB hydrolytic activity. Compared with the vector control, co-expression of either wild-type LpiA or the D750N variant similarly increased the membrane-associated AcvB fraction (Supplementary Fig. 7C, D). Therefore, LpiA enhances AcvB membrane association independently from its catalytic activity. Notably, altering the Lys-PG level did not affect the AcvB membrane association, suggesting that Lys-PG itself did not directly regulate AcvB membrane binding. Taken together, these results indicated that the C-terminal domain of AcvB mediates the interaction with LpiA and that LpiA promotes AcvB membrane association independently from the LpiA catalytic activity and the cellular Lys-PG level.

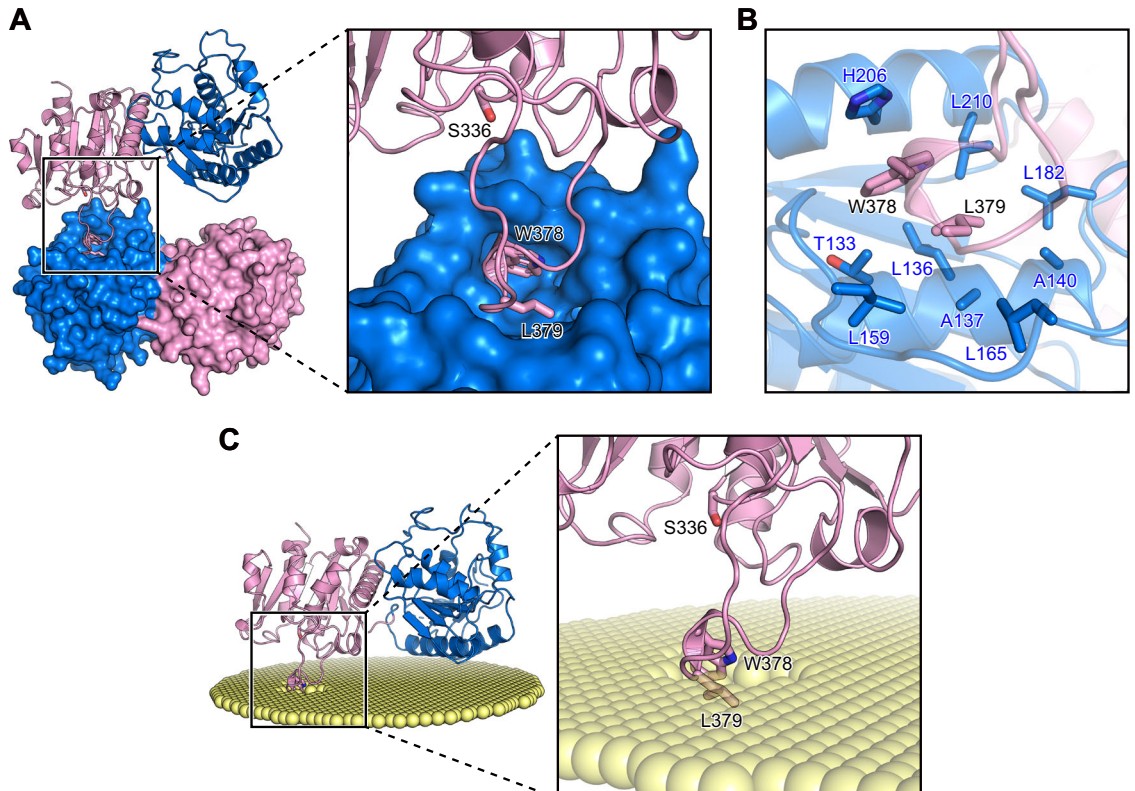

**Fig. 4 | A protruding loop region mediates the membrane association of AcvB.**
**A** Crystal packing interface of AcvB$^{\Delta N24}$ showing the protruding loop in the
C-terminal domain interacting with the N-terminal domain of a neighboring
molecule. Trp378 and Leu379 in the protruding loop, as well as the catalytic residue
Ser336, are shown in stick representation. The neighboring molecule is shown as a
surface representation. **B** Magnified view of the crystal contact shown in (**A**). Trp378

and Leu379 in the protruding loop of one molecule interact with a hydrophobic
pocket in the N-terminal domain of the neighboring molecule. Residues forming the
hydrophobic pocket are shown in stick representation. **C** Predicted membrane-
binding orientation of AcvB$^{\Delta N24}$ calculated using the PPM server. Trp378 and
Leu379 in the protruding loop are positioned to interact with the membrane surface.

## Discussion

In *A. tumefaciens*, AcvB catalyzes the hydrolysis of Lys-PG at the peri-
plasmic side of the inner membrane, thereby maintaining an optimal
balance between Lys-PG synthesis and degradation. In this study, we
determined the crystal structures of the mature form of AcvB (AcvB$^{\Delta N24}$)
and its C-terminal catalytic domain (AcvB$^{C}$) and assessed how the
enzyme recognizes its lipid substrate and interacts with the membrane.
The C-terminal domain forms a negatively charged cavity that accom-
modates the positively charged Lys-PG head group through several acidic
residues, while a hydrophobic protruding loop contributes to membrane
association (Figs. 4 and 5). This loop contains two hydrophobic residues
(Trp378 and Leu379) that are essential for membrane association and
Lys-PG hydrolysis. Notably, the relative orientation of the N- and
C-terminal domains in the crystal structure differed from that predicted
by AlphaFold3[28]. In the predicted model, the protruding loop contacted
the N-terminal domain within the same molecule, whereas in the crystal
structure, it interacted with the hydrophobic region of the N-terminal
domain from a neighboring molecule (Fig. 4A, B). This intermolecular
contact likely mimics the interaction between AcvB and the membrane
surface, suggesting that the domain arrangement observed in the crystal
represents a membrane-bound conformation. Although this interaction
might represent a crystal packing artifact, it raises the possibility that the
N-terminal domain could occlude or restrain the membrane-
interacting loop region in AcvB. Such an arrangement would be con-
sistent with the observed increase in activity upon N-terminal domain
removal (Supplementary Fig. 5). Based on these observations, we spec-
ulate that the N-terminal domain could function as a regulatory module
that modulates catalytic activity, potentially through autoinhibitory
interactions.

Several peripheral enzymes act on membrane phospholipids without
having transmembrane segments. The *E. coli* phosphatidylserine synthase
(PssA), which catalyzes phosphatidylserine synthesis, and the phospholipid
*N*-methyltransferases (PmtA) from *Rhodothermus thermophilus* and *A.
tumefaciens*, which catalyze phosphatidylcholine synthesis, associate with
membranes via amphipathic helices[29–32]. The *E. coli* phosphatidylserine
decarboxylase (Psd), responsible for PE synthesis, binds to membranes via
three hydrophobic helices comprising approximately 60 residues[33,34]. In
contrast, AcvB lacks amphipathic or hydrophobic helices and instead
associates weakly with the membrane via a short loop region. Notably, we
found that the C-terminal domain of AcvB directly interacts with LpiA and
that LpiA promotes AcvB membrane association (Fig. 6A and Supple-
mentary Fig. 7). Because LpiA possesses an integral membrane flippase
domain that translocates Lys-PG from the cytoplasmic to the periplasmic
leaflet, the interaction likely anchors AcvB to the membrane and facilitates
efficient substrate recognition. Therefore, we propose that the LpiA–AcvB
complex constitutes a cooperative module in which LpiA enhances AcvB
membrane localization, thereby promoting efficient substrate recognition.
Although the precise in vivo physiological consequences of LpiA–AcvB
coupling remain elusive, previous work has demonstrated that *acvB* deletion
results yields increased Lys-PG levels and sensitivity to acidic conditions[16].
Moreover, Lys-PG abundance increases under low-pH stress. These
observations suggest that Lys-PG levels need to be tightly regulated during
acid stress. LpiA and AcvB coupling might serve as a mechanism for rapid,
localized, and reversible Lys-PG level modulation in the membrane. Phy-
sical association between the synthase and hydrolase might allow for rapid
Lys-PG abundance adjustment in response to external pH alterations,
thereby helping prevent excessive accumulation while maintaining suffi-
cient levels for stress resistance.

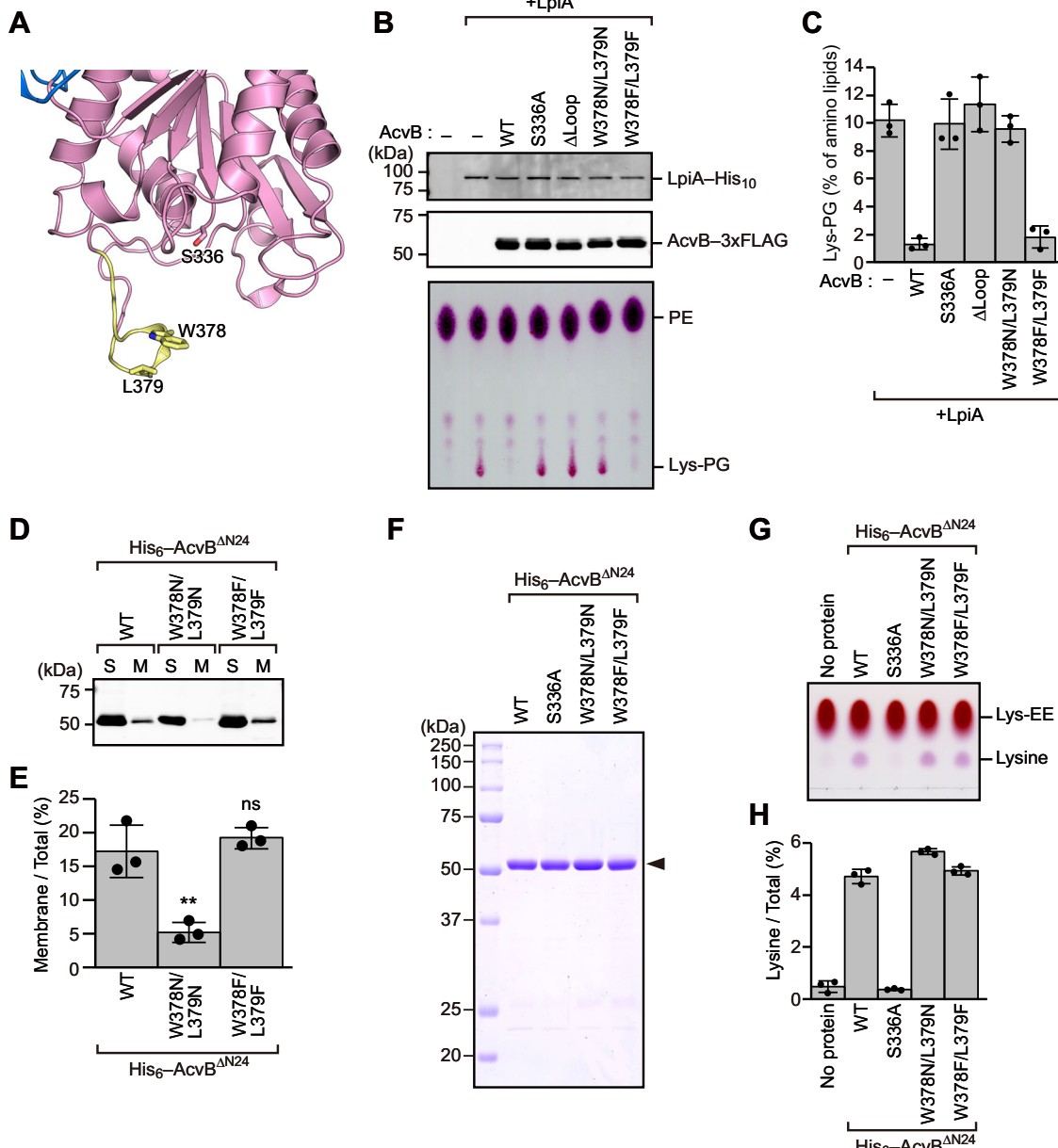

**Fig. 5 | Trp378 and Leu379 contribute to membrane association and Lys-PG hydrolysis. A** Magnified view of the area around the protruding loop region of AcvB[C]. The loop region (residues 373–382) is colored yellow. Trp378 and Leu379 in the protruding loop, as well as the catalytic residue Ser336, are shown in stick representation. **B** *E. coli* cells harboring expression plasmids encoding C-terminally His[10]-tagged LpiA and C-terminally 3xFLAG-tagged AcvB variants were cultured at 37 °C. When the OD[600] reached ~0.5, IPTG was added to a final concentration of 0.5 mM and the cells were incubated at 37 °C for 3 h. Cell lysates were subjected to SDS-PAGE, followed by immunoblotting with anti-6x His antibody and anti-FLAG antibody to detect LpiA and AcvB, respectively. Total phospholipids were extracted, separated by TLC, and visualized using ninhydrin staining. **C** Quantification of Lys-PG based on the data in (**B**), using ImageJ. Values represent mean ± SD from three

independent experiments ($n = 3$). **D** Soluble (S) and membrane (M) fractions were separated from *E. coli* cells expressing N-terminally His[6]-tagged AcvB[ΔN24] variants and subjected to SDS-PAGE and immunoblotting with anti-6x His antibody. **E** Quantification of AcvB in the membrane fraction based on the data in (**D**), using ImageJ. Values represent mean ± SD from three independent experiments ($n = 3$). **, $p < 0.01$, $p$-values were obtained from the unpaired two-tailed t-test. ns not significant. **F** Purified AcvB[ΔN24] variants were analyzed using SDS-PAGE followed by CBB staining. **G** Lys-EE and purified AcvB[ΔN24] variants were incubated at 37 °C for 30 min. Proteins were separated through TLC and visualized using ninhydrin staining. **H** Quantification of lysine based on the data in (**G**), using ImageJ. Values represent mean ± SD from three independent experiments ($n = 3$).

Mutational and docking analyses revealed a potential recognition mechanism for the lysyl head group of Lys-PG by AcvB. Asp271, Asp340, and Asp370 are responsible for Lys-PG hydrolase activity. In the docking model, Asp271 and Asp340 were located close to the α-amino group of the lysine moiety of the substrate, whereas Asp370 was positioned near its ε-amino group. Sequence alignment revealed that Asp271 and Asp340 are conserved in AcvB homologs, including *A. tumefaciens* VirJ, *Rhizobium tropici* AtvA, *B. abortus* VirJ, and the *Pseudomonas*

homologs (PA0919 from *P. aeruginosa* and PP_1201 from *P. putida*) (Supplementary Fig. 8). In contrast, Asp370 is replaced with threonine or serine in *B. abortus* VirJ and the homologs from *P. aeruginosa* and *P. putida*. Given that *A. tumefaciens* and *R. tropici* produce Lys-PG, whereas *P. aeruginosa* and *P. putida* produce Ala-PG, Asp370 likely contributes to the specific recognition of the lysine head group of Lys-PG. Thus, AcvB homologs from bacteria that synthesize Lys-PG may have evolved a more acidic environment around the active site to facilitate

**Fig. 6 | AcvB interacts with LpiA via the C-terminal domain. A** GST pull-down assay showing the interaction between AcvB and LpiA. GST–AcvB$^{ΔN24}$, GST–AcvB$^N$, and GST–AcvB$^C$ bound to GST-accept resin were incubated with LpiA–His$_{10}$. Proteins bound to the resin were eluted with glutathione and analyzed using SDS-PAGE followed by CBB staining and immunoblotting with anti-6x His antibody. **B** Quantification of LpiA–His$_{10}$ in the elution fraction based on the immunoblot data in (**A**), using ImageJ. Values represent mean ± SD from three independent experiments ($n = 3$). *, $p < 0.05$, $p$-values were obtained from the unpaired two-tailed $t$-test. ns not significant. **C** Schematic model illustrating the cooperation between LpiA and AcvB in regulating Lys-PG metabolism in *A. tumefaciens*. LpiA, located in the inner membrane, catalyzes Lys-PG synthesis from PG and lysyl-tRNA and flips Lys-PG to the periplasmic leaflet. AcvB, localized in the periplasm, interacts with LpiA via its C-terminal domain. Multiple acidic residues around the active site recognize the positively charged head group of Lys-PG, whereas a protruding hydrophobic loop contributes to membrane association. AcvB subsequently hydrolyzes Lys-PG to PG and lysine. This figure was created by the authors using Adobe Illustrator.

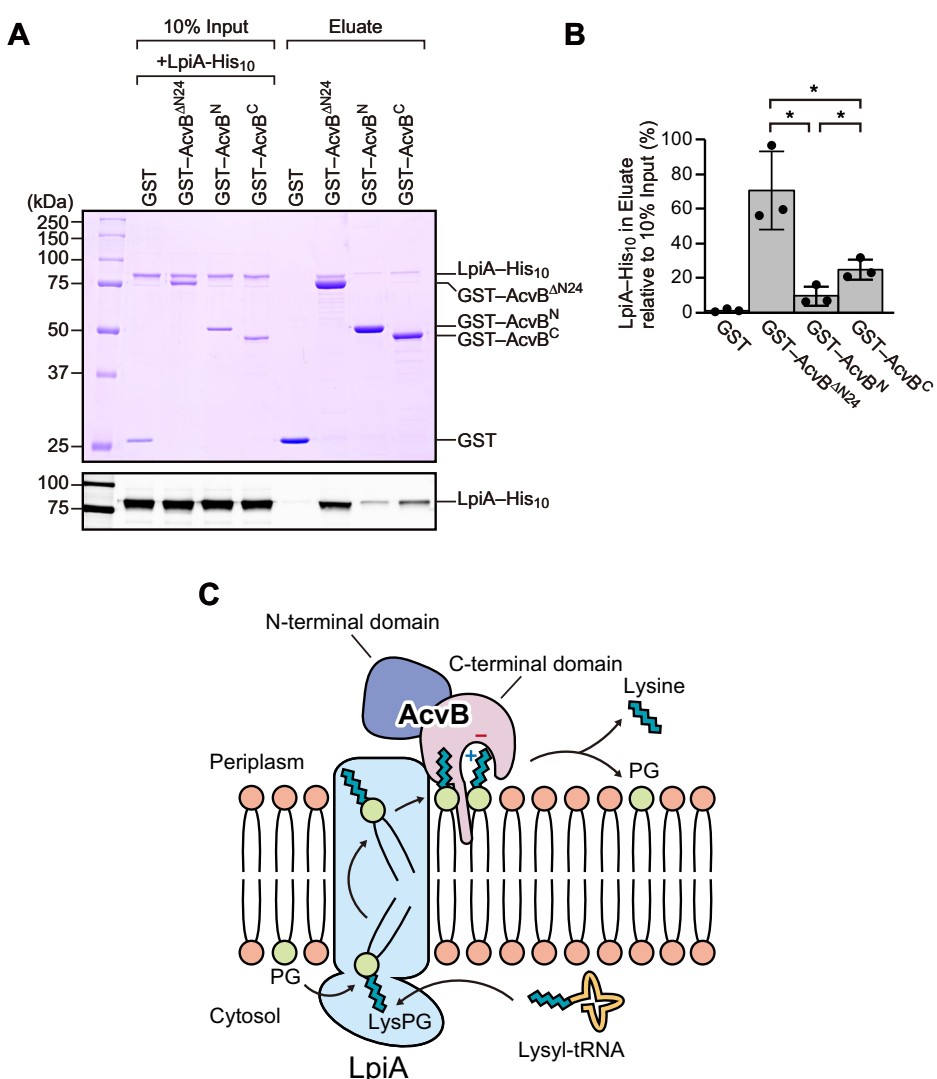

electrostatic complementarity and selective lysine-containing substrate recognition.

Based on the results of structural and biochemical analyses, we propose a working model for Lys-PG hydrolysis by AcvB at the periplasmic membrane surface (Fig. 6C). AcvB associates with the membrane via the Trp378- and Leu379-containing hydrophobic loop, orienting the Ser336-containing catalytic site toward the membrane surface. The lysyl head group of Lys-PG is recognized electrostatically by multiple acidic residues, including Asp271, Asp340, and Asp370, enabling cleavage of the ester linkage. Direct interaction with LpiA via the C-terminal domain of AcvB likely stabilizes the membrane-bound state of AcvB and facilitates efficient substrate transfer between the two enzymes. Through this coupled activity, the LpiA–AcvB module functions to maintain an optimal Lys-PG level and membrane homeostasis in *A. tumefaciens*. Future structural studies of the LpiA–AcvB complex will provide mechanistic insights into substrate transfer between the two enzymes. In conclusion, our findings establish a structural basis for understanding the molecular mechanism of Lys-PG hydrolysis by AcvB and may guide future development of antibacterial agents against plant-pathogenic bacteria.

## Methods
### Construction of expression plasmids
*acvB* and *lpiA* were amplified from the genome of *A. tumefaciens* NBRC 15193. AcvB$^{ΔN24}$ (residues 25–456 of AcvB), AcvB$^N$ (residues 25–248 of

AcvB), and AcvB$^C$ (residues 249–456 of AcvB) were cloned into the pGEX-6p-1 vector (GE Healthcare) to construct N-terminal GST-fusion protein expression plasmids. For FLAG detection, a 3xFLAG tag was additionally introduced at the C-terminus of each construct in the same vector. AcvB$^{ΔN24}$ was also cloned into the pETDuet-1 vector (Novagen) to construct the N-terminally His$_6$-tagged protein expression plasmid. Full-length AcvB was cloned into a modified pACYCDuet-1 vector (Novagen) to add a C-terminal 3xFLAG tag to the expressed protein. LpiA was cloned into a modified pET21a vector (Novagen) to add a C-terminal His$_{10}$-tag to the expressed protein. Mutations for amino-acid substitutions were introduced using PCR-based site-directed mutagenesis. All constructs were sequenced to confirm their identities.

### Protein expression and purification
All constructs were expressed in *E. coli* C43 (DE3) cells (Lucigen) cultured in Luria–Bertani (LB) medium at 37 °C. When the culture reached an optical density at 600 nm (OD$_{600}$) of approximately 0.8, protein expression was induced with 0.1 mM isopropyl β-D-thiogalactopyranoside. After growth at 25 °C for 18 h, the cells were harvested. For GST-tagged AcvB protein purification, cells were resuspended in buffer A (20 mM Tris-HCl [pH 8.0] and 150 mM NaCl) supplemented with 5 mM dithiothreitol, disrupted by sonication, and centrifuged at 20,000×$g$ for 40 min to remove the insoluble debris. The supernatant was loaded onto a GST-Accept column (Nacalai Tesque) equilibrated with buffer A. The column was washed with buffer A,

and GST-tagged proteins were eluted using buffer B (50 mM Tris-HCl [pH 8.0] and 10 mM reduced glutathione). The GST-tag was cleaved using human rhinovirus 3 C protease at 4 °C overnight and removed by reloading the sample onto the GST-Accept column. For N-terminally $His_6$-tagged AcvB protein purification, cells were resuspended in buffer C (50 mM Tris-HCl [pH 8.0], 500 mM NaCl, and 20 mM imidazole). After cell disruption by sonication, the supernatant was loaded onto a Ni–NTA column (Qiagen) equilibrated with buffer C. The column was washed with buffer C, and $His_6$-tagged proteins were eluted using buffer D (50 mM Tris-HCl [pH 8.0], 100 mM NaCl, and 250 mM imidazole). All recombinant AcvB proteins, regardless of the affinity tag used, were further purified through size-exclusion chromatography using a Superdex 200 Increase column (GE Healthcare) with buffer A. GST-tag purified $AcvB^{\Delta N24}$ and $AcvB^C$ proteins were used for crystallization trials after cleavage of the GST-tag.

For C-terminally $His_{10}$-tagged LpiA purification, cells were resuspended in buffer C. After cell disruption by sonication, cell debris was removed by centrifugation at $10,000 \times g$ for 10 min and the membrane fraction was collected by ultracentrifugation at $100,000 \times g$ for 90 min. The membrane fraction was solubilized in buffer C supplemented with 1.5% dodecyl-β-D-maltopyranoside (DDM) at 4 °C for 60 min. Insoluble components were removed by ultracentrifugation at $100,000 \times g$ for 30 min and the supernatant was loaded onto a Ni–NTA column equilibrated with buffer C supplemented with 0.03% DDM. The column was washed with buffer C supplemented with 0.03% DDM, and the protein was eluted using buffer D supplemented with 0.03% DDM. LpiA was further purified through size-exclusion chromatography using a Superdex 200 Increase column with buffer A supplemented with 0.03% DDM.

## Crystallization and X-ray crystallography

Crystallization trials were performed at 20 °C using the sitting drop vapor diffusion method. For crystallization of $AcvB^{\Delta N24}$, 0.2-μL drops of approximately 27 mg/mL $AcvB^{\Delta N24}$ in 20 mM Tris-HCl (pH 8.0) and 150 mM NaCl were mixed with an equal amount of reservoir solution consisting of 2.4 M sodium malonate (pH 7.0) and equilibrated against 70 μL of the same reservoir solution through vapor diffusion. For crystallization of $AcvB^C$, 0.2-μL drops of approximately 6.8 mg/mL $AcvB^C$ in 20 mM Tris-HCl (pH 8.0) and 150 mM NaCl were mixed with an equal amount of reservoir solution consisting of 0.1 M ammonium iodide, 0.1 M sodium acetate (pH 6.0), and 20% $w/v$ polyethylene glycol 3350, and equilibrated against 70 μL of the same reservoir solution through vapor diffusion. The crystals were soaked in a reservoir solution supplemented with 15% ethylene glycol, flash-cooled, and maintained in a stream of nitrogen ($N_2$) gas at 100 K during data collection. X-ray diffraction data were collected at the SPring-8 beamline BL32XU, with a $10 \times 15$-μm (width × height) microbeam using the helical data collection method. The diffraction data were collected using the ZOO automated data collection system[35]. The data were processed using the KAMO[36] and XDS[37] software. Structures were determined through molecular replacement with PHASER[38], using the structure predicted by ColabFold[23,24] as a search model. Further model building was performed manually using COOT[39], and crystallographic refinement was performed using PHENIX[40]. MolProbity[41] was used to assess the quality and geometry of structural models. Detailed data collection and processing statistics are shown in Table 1. Ramachandran analysis showed that 96.64% and 97.88% of $AcvB^{\Delta N24}$ and $AcvB^C$ residues were located in favored regions, and 3.36% and 2.12% in allowed regions, respectively, whereas no outliers were observed.

## Molecular docking

Molecular docking of a lysyl-glycerol molecule into the crystal structure of $AcvB^C$ was performed using the SwissDock server[42], which is based on AutoDock Vina[25]. Prior to docking, all water molecules were removed from the $AcvB^C$ structure. Docking was performed with a sampling exhaustivity of 8. The $AcvB^C$ structure was set as rigid during docking, and a $30 \times 30 \times 30$ Å grid box was placed around the S336 residue. The top-ranked docking pose showed a binding affinity of −5.34 kcal/mol. The predicted binding mode was consistent with the positions of conserved acidic residues and with mutational analysis.

## Separation of soluble and membrane fractions

Soluble and membrane fractions of E. coli lysates were separated as previously described[32,33], with several modifications. E. coli C43(DE3) cells carrying expression plasmids for C-terminally $His_{10}$-tagged LpiA and C-terminally 3xFLAG-tagged AcvB variants were cultured in 80 mL of LB medium at 37 °C. When the $OD_{600}$ reached approximately 0.5, isopropyl β-D-1-thiogalactopyranoside (IPTG) was added to a final concentration of 500 μM, and the cultures were incubated at 37 °C for 3 h to induce protein expression. Next, the cells were harvested, resuspended in lysis buffer (20 mM Tris-HCl [pH 8.0] and 150 mM NaCl), disrupted through sonication, and centrifuged at $10,000 \times g$ for 10 min to pellet the insoluble debris. The samples were ultracentrifuged at $150,000 \times g$ for 1 h. The supernatant and pellet fractions were defined as soluble and membrane fractions, respectively, and analyzed using sodium dodecyl sulfate-polyacrylamide gel electrophoresis (SDS-PAGE) followed by immunoblotting.

## TLC analysis

To analyze the hydrolysis of Lys-PG by AcvB variants expressed in E. coli cells, membrane fractions were prepared from 80-mL E. coli cultures and resuspended in 100 μL of lysis buffer. Fifty microliters of the membrane suspension was mixed with 750 μL of chloroform/methanol (2:1, $v/v$) and vortexed for 10 min. Then, 100 μL of water was added, and the samples were vortexed again for 10 min. The organic phase was separated by centrifugation at $1000 \times g$ for 2 min, collected, and dried under a stream of $N_2$ gas. The resulting lipid films were dissolved in 40 μL of chloroform. Twenty-microliter aliquots of each sample were spotted onto an HPTLC Silica gel 60 plate (Merck Millipore) and analyzed through TLC using chloroform/methanol/water (65:25:4, $v/v/v$).

To analyze the hydrolysis of Lys-EE by purified AcvB variants, 40 mM Lys-EE was incubated with 40 μM purified AcvB variants in 50 μL of assay buffer A (8 mM Tris-HCl [pH 8.0] and 60 mM NaCl) at 37 °C for 30 min. The reaction was stopped by adding 5 μL of 10% SDS solution. One-microliter aliquots of each sample were spotted onto an HPTLC Silica gel 60 plate and analyzed through TLC using 1-butanol/acetic acid/water (3:1:1, $v/v/v$).

To analyze Lys-PG hydrolysis by purified AcvB variants, 0.2 mM of 18:1-18:1 Lys-PG (Avanti Polar Lipids) was incubated with 20 μM purified AcvB variants in 50 μL of assay buffer B (20 mM HEPES [pH 7.0], 150 mM NaCl, and 0.2% Triton X-100) at 37 °C. After incubation for the indicated time periods, 750 μL of chloroform/methanol (2:1, $v/v$) was added, and the samples were vortexed for 10 min. After the addition of 100 μL of water, the samples were vortexed again for 10 min. The organic phase was separated by centrifugation at $1000 \times g$ for 2 min, collected, and dried under a stream of $N_2$ gas. The resulting lipid films were dissolved in 40 μL of chloroform. Twenty-microliter aliquots of each sample were spotted onto a TLC Silica gel 60 plate and analyzed through TLC using chloroform/methanol/water (65:25:4, $v/v/v$). Sample spots were visualized through staining with ninhydrin using Ninhydrin–ethanol TS Spray (FUJIFILM Wako).

## In vitro pull-down assay

For the GST pull-down assay, GST-tagged AcvB variants (10 μM) and C-terminally $His_{10}$-tagged LpiA (10 μM) were incubated with 50 μL of GST-Accept resin in buffer containing 20 mM Tris-HCl (pH 8.0), 150 mM NaCl, and 0.03% DDM at 4 °C for 60 min. The resin was washed three times with 500 μL of the same buffer, and bound proteins were eluted with 150 μL of 10 mM glutathione in 50 mM Tris-HCl (pH 8.0) and 0.03% DDM. Input samples (mixtures of GST-tagged AcvB variants and $His_{10}$-tagged LpiA before incubation with the resin) and eluted fractions were analyzed using SDS-PAGE followed by Coomassie brilliant blue (CBB) staining and immunoblotting with an anti-6x His antibody.

For the Ni-NTA pull-down assay, C-terminally 3xFLAG-tagged AcvB variants (10 μM) and C-terminally His$_{10}$-tagged LpiA (10 μM) were incubated with 50 μL of Ni–NTA resin in buffer containing 20 mM Tris-HCl (pH 8.0), 150 mM NaCl, and 0.03% DDM at 4 °C for 30 min. The resin was washed three times with 500 μL of wash buffer (50 mM Tris-HCl (pH 8.0), 150 mM NaCl, 20 mM imidazole, and 0.03% DDM), and bound proteins were eluted with 200 μL of elution buffer (50 mM Tris-HCl (pH 8.0), 100 mM NaCl, 250 mM imidazole, and 0.03% DDM). Input samples (mixtures of C-terminally 3xFLAG-tagged AcvB variants and His$_{10}$-tagged LpiA before incubation with the resin) and eluted fractions were analyzed using SDS-PAGE followed by CBB staining and immunoblotting with an anti-FLAG antibody.

## Immunoblotting

For immunoblotting, mouse monoclonal anti-6x His antibody (clone 9C11; FUJIFILM Wako) and mouse monoclonal anti-FLAG antibody (clone M2; Sigma-Aldrich) were used at dilutions of 1:2000 and 1:3000, respectively. A fluorophore-conjugated secondary antibody, goat anti-mouse IgG (H + L) cross-absorbed secondary antibody, cyanine5 (A10524; Life Technologies), was used at a 1:2000 dilution. Fluorescent signals were detected using a Typhoon FLA 9500 imaging system (GE Healthcare).

## Amino-acid sequence alignment

The amino acid sequences of AcvB and its homologs were analyzed using Clustal Omega[43] and ESPript 3.0[44].

## Statistics and reproducibility

Data in Figs. 2E, 3E, 5C, E, H, and 6B, and Supplementary Figs. 5B and 7D were obtained from three independent experiments and are presented as the mean ± standard deviation (SD). Representative TLC and western blot images from these experiments are presented in Figs. 2D, 3D, 5B, D, G, and 6A as well as in Supplementary Figs. 5A and 7C. Quantification was performed using ImageJ software.

## Reporting summary

Further information on research design is available in the Nature Portfolio Reporting Summary linked to this article.

## Data availability

The atomic coordinates and structure factor files were deposited in the Protein Data Bank under accession codes 9XHM (AcvB$^{ΔN24}$) and 9XHN (AcvB$^C$). Uncropped images of blots, gels, and TLC plates are provided in Supplementary Fig. 9. Source data for graphs are available in Supplementary Data 1. All other data are available from the corresponding author on reasonable request.

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

## Acknowledgements

We thank Prof. Yasushi Tamura for his valuable advice, insightful discussions, and support with experimental instruments. This work was supported by the Japan Society for the Promotion of Science KAKENHI (Grant Nos. JP25K09522 and JP22K06096 to Y.W.), and Institute for Fermentation, Osaka (to Y.W.). Synchrotron radiation experiments were performed at beamline BL32XU at SPring-8, Japan, with approval from the Japan Synchrotron Radiation Research Institute (Proposal Nos. 2024B2757, 2024A2757, 2023B2762, 2023A2762, and 2022A2760).

## Author contributions

Y.W. conceived the project. M.H. performed most of the experiments. M.H., D.M., and Y.W. performed the sample preparation. M.H. and Y.W. conducted structural studies. Y.W. wrote the original draft, and all authors reviewed and edited the manuscript.

## Competing interests

The authors declare no competing interests.
