## [Transparent Peer Review file · Communications Biology]

Structural basis of substrate recognition and membrane association by the bacterial lysyl-phosphatidylglycerol hydrolase AcvB

Corresponding Author: Dr Yasunori Watanabe

Version 0:

Reviewer comments:

Reviewer #1

(Remarks to the Author)

Review of the manuscript entitled Structural basis of substrate recognition and membrane association by the bacterial lysyl-phosphatidylglycerol hydrolase AcvB by Hoshi and Watanabe

The article describes a structural and biochemical study of the AcvB protein from *Agrobacterium tumefaciens*, a periplasmic enzyme that hydrolyzes L-PG into PG. The authors have solved the crystal structure of the full length protein and of the catalytic domain (C-terminal) at resolutions of 3.1 and 1.8 Å, respectively.

These structures reveal that AcvB contains two esterase folds (N- and C-domains). Interestingly, the AlphaFold prediction of the domains was correct but not the overall conformation of the protein, showing that experimental structures are still essential for our understanding of enzymes.

Based on the structure, the authors derive a number of experiments : mutagenesis, docking, enzymatic assays, and protein interaction studies, that successfully identify the active site and residues involved in hydrolysis or in recognizing the substrate. They do this through multiple enzymatic assays and co expression with the LpiA enzyme. Based on a crystallization artefact, they identify a protruding hydrophilic loop that is demonstrated to be involved in interacting with the membrane.

Finally, they identify the C-terminal domain of AcvB as interacting with the LpiA protein.

Based on all their results, the authors propose a model of function and association between AcvB and LpiA at the outer leaflet of the inner membrane.

Overall, I found the methodology sound, the experimental design very good, and the evidence compelling. The structural work is excellent and the analysis very well carried out. The figures are very well designed.

I have very few comments on this manuscript.

Major

1. It is unclear what the benefit is of the physical association of LpiA and AcvB. Indeed, if the bacterium produces L-PG with LpiA, why couple it to its hydrolysis by AcvB? If this coupling is strict, then L-PG would never be present in the membrane. Can the authors comment?
2. The authors show that the C-terminal domain of AcvB hydrolyzes L-PG, interacts with the membrane, and interacts with LpiA. What is the role of the N-terminal domain? Can they speculate ?
3. Can the authors provide the reviewers with PDB validation reports of their crystal structures?
4. The authors mention in the introduction the crystal structure of the domain D1 of an AcvB homologue from *Brucella*. Can they compare this structure ? Is domain D1 similar to one of AcvB ?

Minor

• Abstract:

line 4 : LpiA and... AcvB genes. LpiA and AcvB should not have capital letters and should be italicized when referring to

genes.

line 9 : mediates transient membrane association.

Why do the authors think the interaction is transient ?

• Introduction:

line 33 : the sentence decreasing in their membrane does not make sense; likely a typo.

In the manuscript the authors use AcvB(25–456) to describe their protein. They use additional labeling for mutations, resulting in names that are rather long, e.g., AcvB(25–456)(S336A) in line 76 of the Results section (for instance). Could they use another notation? Maybe AcvB for the periplasmic form, and then AcvBN or AcvBC with superscripts? This is just a suggestion.

Reviewer #2

(Remarks to the Author)

This straightforward manuscript presents the first in-depth structural analysis of a bacterial lysyl-PG hydrolase. The topic is timely and relevant, as the balance between PG and lysyl-PG influences the net charge of the membrane and thereby affects diverse cellular processes, including antibiotic resistance and virulence in both mammalian and plant pathogens.

Hoshi and Watanabe successfully crystallized the mature form of AcvB (full-length protein lacking the signal peptide) at moderate resolution (3.1 Å), as well as the catalytic C-terminal domain at higher resolution (1.8 Å). They further explored lysyl-PG binding, membrane association and interaction with the lysyl-PG-forming enzyme LpiA using site-directed mutagenesis, computational modeling and pulldown experiments.

The manuscript is logically structured and clearly written. Most of the results are convincing and well documented. I have only a few questions and suggestions for improvement.

Major issues:

1. The interaction study between AcvB and LpiA (Fig. 6) is the weakest part of the manuscript. The interaction appears far from stoichiometric, as AcvB bands are clearly visible on Coomassie gels, whereas LpiA is undetectable without Western blotting.

(a) The experimental protocol needs a more detailed description. The lane labeling is confusing, for example, lane 1 is labeled "GST" but appears to contain GST and LpiA. Please clarify.

(b) Some residual LpiA was detected in lanes 1 and 3, yet the authors state that the N-terminal AcvB "did not bind". How often was this experiment reproduced, and were the results consistent and statistically significant?

(c) Ideally, the authors should perform a reverse pulldown assay by immobilizing the His-tagged protein on a Ni-NTA column and incubating it with the GST fusions.

(d) Can the interaction between LpiA and the C-terminal AcvB domain be supported by AlphaFold-mediated prediction?

2. It is plausible that LpiA and AcvB function together as "module to maintain an optimal lysyl-PG level and membrane homeostasis". However, this model raises an important question: How is a futile cycle avoided in which freshly flipped amino-acylated PG is immediately hydrolyzed back to PG? One possibility is that the transient interaction of AcvB with the membrane and/or LpiA is regulated by the Lys-PG/PG ratio. Do the authors have any hypotheses or mechanistic ideas how such regulation might be achieved?

Minor issues:

1. Protein nomenclature is currently awkward and confusing. The manuscript uses multiple designations for the mature form of AcvB, such as AcvB(25-456) or wild-type protein (Supp. Fig. 1). It gets more complicated for mutated variants, e.g. "S336A-mutant AcvB(25-456)". Please harmonize the nomenclature early in the manuscript, for example by defining on page 3, line 58 that the mature form of the protein (lacking the signal sequence) will be referred to simply as AcvB throughout the text.

2. Carefully check gene versus protein designations and the use of the term "mutant". A mutant refers to an organism, not to a specific gene or protein. A protein carrying an amino acid substitution should not be described as a mutant.

3. Page 2, line 33: What do the authors mean by "decreasing in their membrane"?

4. Page 4, line 118: The statement that "E. coli cells harboring empty vectors contained only PE" is misleading. Because the separated lipid extracts were stained with ninhydrin, only nitrogen-containing lipids are visualized, and PC and CL would not be detected under these conditions. The same issue applies to figures labeled as showing "total lipids [%]". These panels actually represent "[%] of amino lipids" and not total lipid composition, if I am not mistaken.

5. Page 6, line 189: A "not" is missing.

6. Page 7, line 243: To the best of my knowledge, the membrane association of PmtA via amphipathic helices was first shown by Danne et al. (2015).

7. In the Introduction, the authors mention a paper on the N-terminal D1 domain structure of the *Brucella abortus* AcvB homolog VirJ (reference 20). It would be worthwhile to revisit this study in the Discussion and compare those findings with the result presented here. *Brucella* VirJ is not included in the sequence alignment in Fig. S4, and it is unclear where domain 1 of the *Brucella* protein is located compared to AcvB.

Reviewer #3

(Remarks to the Author)

Agrobacterium chromosomal virulence protein B (AcvB) is a periplasmic hydrolase involved in maintaining the level of lysyl-phosphatidylglycerol (Lys-PG) by catalyzing hydrolysis of Lys-PG into PG and lysine. The function of AcvB is crucial for the bacterial cells (*A. tumefaciens*) to adapt to acidic conditions and achieve optimal virulence. In the manuscript presented by

Hoshi, M. and Watanabe Y., the crystal structures of mature AcvB (full length form lacking the signal sequence) and the C-terminal catalytic domain have been solved. The authors have identified a potential binding site for the Lys-PG head group in a negatively-charged cavity of the catalytic domain, analyzed the activities of a series of site-directed mutants in comparison with the wild type enzyme and detected a potential interaction between AcvB and LpiA. The results are new and beneficial for understanding the molecular mechanism underlying the catalytic function of AcvB.

The following are a few questions/comments for the authors to consider for further revision and improvement of their work.

Major points:

1) In lines 85-87, the authors described the overall fold of the N-terminal domain of AcvB. What is the function of this domain? Does it also have the LysPG (or other substrate) hydrolyzing activity as the C-terminal domain, participate in substrate binding or regulate the catalytic activity of the C-terminal domain?

2) In lines 100-101, it was stated that the C-terminal domain of AcvB is necessary and sufficient for the Lys-PG hydrolase activity. How does the specific activity of the recombinant C-terminal domain of AcvB compare to the mature AcvB protein (with both N-terminal and C-terminal domains)? The comparison may also help to address the above question about whether the N-terminal domain has a role in regulating the catalytic activity of the C-terminal domain.

3) In line 207-208, it was claimed that Trp378 and Leu379 contribute to the recognition of the acyl chains of Lys-PG through their hydrophobic side chains. However, I could not find any direct evidences supporting the conclusion. The data presented in Figure 5 only demonstrates that Trp378 and Leu379 are crucial for the activity of AcvB and association of the protein with membrane, but do not necessarily indicate that they are involved in recognizing the acyl chains of Lys-PG. To provide direct evidences, the authors could apply the protein lipid overlay assay (Dowler et al., Sci STKE 129:pl6, 2002) to analyze the interactions between AcvB (WT & mutant) and Lys-PG (and other lipids with different head group and acyl chain compositions).

4) The interaction between AcvB and LpiA discovered through the GST pull-down assays is very interesting. It will be great if the authors could measure the kinetic parameters of their interactions by using one of the quantitative methods, such as ITC, SPR or BLI.

Minor points:

1) In Figure 3A, the distances (Å) between two hydrogen-bonded groups should be measured and labeled nearby the dash lines.

2) In the legends of Figures 2E, 3E, 5C, 5E, 5H, it was stated that the histograms are the ImageJ quantification results based on the data presented in Fig. 2D, 3D, 5B, 5D and 5G respectively. As there are three repeats for each data point, please clarify if the repeats are repeated measurements on the same TLC/western blot image or multiple repeats of the TLC/western blot experiments with one representative image shown in the figure.

3) The gap between Rwork and Rfree for the structure of AcvB (25-456) is a bit too large (Table 1), maybe due to overfitting. The RMS deviation values of bond lengths and bond angles appear to be fairly low, as the stereochemistry might be too tightly restrained. The target weights could be optimized further to decrease the Rfree or reduce the gap between Rwork and Rfree. It will be better to reduce the gap to around 3% if possible.

4) There are several typos in the manuscript to be fixed, such as "decreasingin their membrane" (line 33-34) and "whereas the W378F/L379F mutant did (not)" (line 189). Please check the manuscript carefully to minimize typo errors.

Version 1:

Reviewer comments:

Reviewer #1

(Remarks to the Author)

The authors have successfully addressed all my comments, provided additional information and ave modified the manuscript extensively.

I believe I have no further modification or question on this very nice study.

Reviewer #2

(Remarks to the Author)

The authors responded to all reviewer comments, performed additional experiments and adapted the text accordingly. I am satisfied with the revision.

Reviewer #3

(Remarks to the Author)

The authors have addressed my previous questions/comments constructively or in a reasonable way. The manuscript has

been improved after revision. I have no further questions.

Response to the Reviewer #1

Review of the manuscript entitled Structural basis of substrate recognition and membrane association by the bacterial lysyl-phosphatidylglycerol hydrolase AcvB by Hoshi and Watanabe

The article describes a structural and biochemical study of the AcvB protein from Agrobacterium tumefaciens, a periplasmic enzyme that hydrolyzes L-PG into PG. The authors have solved the crystal structure of the full length protein and of the catalytic domain (C-terminal) at resolutions of 3.1 and 1.8 Å, respectively.

These structures reveal that AcvB contains two esterase folds (N- and C-domains). Interestingly, the AlphaFold prediction of the domains was correct but not the overall conformation of the protein, showing that experimental structures are still essential for our understanding of enzymes.

Based on the structure, the authors derive a number of experiments : mutagenesis, docking, enzymatic assays, and protein interaction studies, that successfully identify the active site and residues involved in hydrolysis or in recognizing the substrate. They do this through multiple enzymatic assays and co expression with the LpiA enzyme. Based on a crystallization artefact, they identify a protruding hydrophilic loop that is demonstrated to be involved in interacting with the membrane.

Finally, they identify the C-terminal domain of AcvB as interacting with the LpiA protein.

Based on all their results, the authors propose a model of function and association between AcvB and LpiA at the outer leaflet of the inner membrane.

Overall, I found the methodology sound, the experimental design very good, and the evidence compelling. The structural work is excellent and the analysis very well carried out. The figures are very well designed.

I have very few comments on this manuscript.

We are grateful to the Reviewer 1 for the positive comments on our manuscript.

Major

1. It is unclear what the benefit is of the physical association of LpiA and AcvB. Indeed, if the bacterium produces L-PG with LpiA, why couple it to its hydrolysis by AcvB? If this coupling is strict, then L-PG would never be present in the membrane. Can the authors comment?

We thank the reviewer for raising this important point. Although our study does not directly address the in vivo physiological consequences of LpiA–AcvB coupling, previous work has shown that deletion of *acvB* results in elevated Lys-PG levels and increased sensitivity to acidic conditions (Groenewold *et al.*, 2019). In addition, Lys-PG abundance increases under low-pH stress. These findings indicate that Lys-PG levels need to be tightly regulated during acid stress. Rather than constituting a futile cycle, LpiA and AcvB coupling might enable rapid, localized, and reversible modulation of Lys-PG levels in the membrane. Physical association between the synthase and hydrolase might allow rapid adjustment of Lys-PG abundance in response to changes in external pH, thereby preventing excessive accumulation while maintaining sufficient

levels for stress resistance. We have added this explanation to the main text of the revised manuscript (lines 282–290).

[Reference]

Groenewold, M. K. *et al.* Virulence of *Agrobacterium tumefaciens* requires lipid homeostasis mediated by the lysyl-phosphatidylglycerol hydrolase AcvB. *Mol Microbiol* **111**, 269–286 (2019).

2. The authors show that the C-terminal domain of AcvB hydrolyzes L-PG, interacts with the membrane, and interacts with LpiA. What is the role of the N-terminal domain? Can they speculate?

We appreciate the reviewer's insightful question. We examined the Lys-PG hydrolase activities of the mature form of AcvB, its N-terminal domain, and its C-terminal domain (Supplementary Fig. 5 in revised manuscript). We have added this analysis to the main text of the revised manuscript (lines 104–109). Consistent with the previous study, the C-terminal domain exhibited Lys-PG hydrolase activity, whereas the N-terminal domain alone did not show detectable activity. Interestingly, the C-terminal domain displayed higher hydrolase activity than the mature form of AcvB, suggesting that the N-terminal domain may negatively regulate catalytic activity. In the crystal structure, the membrane-binding hydrophobic loop of the C-terminal domain interacts with the N-terminal domain of a neighboring AcvB molecule (Fig. 4A, B). Although this interaction might represent a crystal packing artifact, it raises the possibility that the N-terminal domain could occlude or restrain the membrane-interacting region in the mature form of AcvB. Such an arrangement would be consistent with the observed increase in activity upon N-terminal domain removal. Based on these observations, we speculate that the N-terminal domain could function as a regulatory module that modulates catalytic activity, potentially through autoinhibitory interactions. We have added this speculation to the main text of the revised manuscript (lines 261–267).

3. Can the authors provide the reviewers with PDB validation reports of their crystal structures?

We have attached PDB validation reports of crystal structures.

4. The authors mention in the introduction the crystal structure of the domain D1 of an AcvB homologue from Brucella. Can they compare this structure? Is domain D1 similar to one of AcvB?

We appreciate the reviewer's comment. We compared the N-terminal domain of AcvB with the D1 domain of *B. abortus* VirJ and found structural similarity (r.m.s.d. 2.32 Å over 145 Ca atoms). We have described this comparison to the Results section (lines 89 – 91) and added Supplementary Fig. 3 to the revised manuscript.

Minor

• *Abstract:*

line 4 : LpiA and... AcvB genes. LpiA and AcvB should not have capital letters and should be italicized when referring to genes.

We have corrected the gene names to lowercase italics in the revised manuscript.

line 9 : mediates transient membrane association.

Why do the authors think the interaction is transient ?

We thank the reviewer for this comment. To avoid overinterpretation, we have removed the term “transient” in the revised manuscript.

• *Introduction:*

line 33 : the sentence decreasing in their membrane does not make sense; likely a typo.

We have corrected the text in the revised manuscript.

In the manuscript the authors use AcvB(25–456) to describe their protein. They use additional labeling for mutations, resulting in names that are rather long, e.g., AcvB(25–456)(S336A) in line 76 of the Results section (for instance). Could they use another notation? Maybe AcvB for the periplasmic form, and then AcvBN or AcvBC with superscripts? This is just a suggestion.

We thank the reviewer for this comment. In the revised manuscript, we have referred to AcvB(25–456), AcvB(25–248), and AcvB(249–456) as AcvB^{ΔN24}, AcvB^N, and AcvB^C, respectively.

Response to the Reviewer #2

This straightforward manuscript presents the first in-depth structural analysis of a bacterial lysyl-PG hydrolase. The topic is timely and relevant, as the balance between PG and lysyl-PG influences the net charge of the membrane and thereby affects diverse cellular processes, including antibiotic resistance and virulence in both mammalian and plant pathogens.

Hoshi and Watanabe successfully crystallized the mature form of AcvB (full-length protein lacking the signal peptide) at moderate resolution (3.1 Å), as well as the catalytic C-terminal domain at higher resolution (1.8 Å). They further explored lysyl-PG binding, membrane association and interaction with the lysyl-PG-forming enzyme LpiA using site-directed mutagenesis, computational modeling and pulldown experiments.

The manuscript is logically structured and clearly written. Most of the results are convincing and well documented. I have only a few questions and suggestions for improvement.

We are grateful to the Reviewer 2 for the positive comments on our manuscript.

Major issues:

1. The interaction study between AcvB and LpiA (Fig. 6) is the weakest part of the manuscript. The interaction appears far from stoichiometric, as AcvB bands are clearly visible on Coomassie gels, whereas LpiA is undetectable without Western blotting.

(a) The experimental protocol needs a more detailed description. The lane labeling is confusing, for example, lane 1 is labeled “GST” but appears to contain GST and LpiA. Please clarify.

We thank the reviewer for this comment. We have revised the Methods section to provide a more detailed description of the in vitro pull-down assay protocol (lines 442–444). In addition, the lane

labels in Fig. 6A have been clarified to avoid confusion, and the figure has been updated accordingly.

(b) Some residual LpiA was detected in lanes 1 and 3, yet the authors state that the N-terminal AcvB “did not bind”. How often was this experiment reproduced, and were the results consistent and statistically significant?

We thank the reviewer for this comment. We repeated the GST pull-down assay three times using freshly purified GST-tagged AcvB variants and LpiA–His₁₀. The results were consistent across replicates. Quantification of the co-eluted LpiA (Fig. 6A, B in the revised manuscript) showed that the C-terminal domain of AcvB interacted with LpiA, although the interaction was weaker than that observed for AcvB^{ΔN24}. In contrast, the N-terminal domain exhibited only a weak association with LpiA compared with both AcvB^{ΔN24} and the C-terminal domain of AcvB. We have revised the Results section accordingly (lines 226–232).

(c) Ideally, the authors should perform a reverse pulldown assay by immobilizing the His-tagged protein on a Ni-NTA column and incubating it with the GST fusions.

We appreciate the reviewer for this suggestion. We performed a reciprocal Ni-NTA pull-down assay using freshly purified FLAG-tagged AcvB variants and LpiA–His₁₀. In this assay, His₁₀-tagged LpiA immobilized on Ni-NTA resin pulled down AcvB^{ΔN24} and the C-terminal domain of AcvB but not the N-terminal domain of AcvB (Supplementary Fig. 6 in the revised manuscript). These results indicated that the C-terminal domain predominantly mediates the interaction with LpiA, while the contribution of the N-terminal domain might be weak and assay-dependent. We have revised the Results section accordingly (lines 226–232).

(d) Can the interaction between LpiA and the C-terminal AcvB domain be supported by AlphaFold-mediated prediction?

We attempted to predict the structure of the LpiA–AcvB complex using AlphaFold3. In the predicted model, the N-terminal domain of AcvB was positioned near the cytosolic domain of LpiA (Figure A). However, this arrangement is unlikely to represent the physiological interaction because AcvB is localized in the periplasm, whereas the predicted interface involves the cytosolic region of LpiA. Furthermore, the predicted aligned error (PAE) between LpiA and AcvB was high, indicating low confidence in the relative positioning of the two proteins (Figure B). Therefore, the AlphaFold3 prediction does not provide reliable structural support for the interaction observed in our biochemical assays.

Figure only for reviewers.

(A) AlphaFold3-predicted structure of the LpiA-AcvB complex.

(B) Predicted aligned error (PAE) map for the LpiA-AcvB complex predicted by AlphaFold3.

2. It is plausible that LpiA and AcvB function together as “module to maintain an optimal lysyl-PG level and membrane homeostasis”. However, this model raises an important question: How is a futile cycle avoided in which freshly flipped amino-acylated PG is immediately hydrolyzed back to PG? One possibility is that the transient interaction of AcvB with the membrane and/or LpiA is regulated by the Lys-PG/PG ratio. Do the authors have any hypotheses or mechanistic ideas how such regulation might be achieved?

We thank the reviewer for this insightful conceptual question. To examine whether AcvB membrane association depends on Lys-PG levels in the membrane, we co-expressed a catalytically inactive AcvB(S336A) variant with either wild-type LpiA or the catalytically inactive LpiA(D750N) variant. In both cases, co-expression of LpiA increased the membrane-associated fraction of AcvB to a similar extent (Supplementary Fig. 7). We have added this analysis to the main text of the revised manuscript (lines 233–244). These data indicate that LpiA enhances AcvB membrane association independently from the LpiA catalytic activity and the cellular Lys-PG level. These findings argue against a simple model in which the Lys-PG/PG ratio directly regulates AcvB localization. We therefore consider it likely that additional regulatory factors beyond Lys-PG abundance are involved in preventing futile cycling. However, elucidating the precise mechanism will require further investigation and is beyond the scope of the present study.

Minor issues:

1. Protein nomenclature is currently awkward and confusing. The manuscript uses multiple designations for the mature form of AcvB, such as AcvB(25-456) or wild-type protein (Supp. Fig. 1). It gets more complicated for mutated variants, e.g. “S336A-mutant AcvB(25-456)”. Please harmonize the nomenclature early in the manuscript, for example by defining on page 3, line 58 that the mature form of the protein (lacking the signal sequence) will be referred to simply as AcvB throughout the text.

We thank the reviewer for this comment. To harmonize the nomenclature, we simplified the

designation of AcvB constructs throughout the revised manuscript. AcvB(25–456), AcvB(25–248), and AcvB(249–456) are now referred to as AcvB^{ΔN24}, AcvB^N, and AcvB^C, respectively.

2. Carefully check gene versus protein designations and the use of the term “mutant”. A mutant refers to an organism, not to a specific gene or protein. A protein carrying an amino acid substitution should not be described as a mutant.

We thank the reviewer for this comment. We revised gene and protein designations throughout the manuscript and replaced the term “mutant” with “variant” or “substitution” where appropriate.

3. Page 2, line 33: What do the authors mean by “decreasing in their membrane”?

We have corrected the text in the revised manuscript.

4. Page 4, line 118: The statement that “E. coli cells harboring empty vectors contained only PE” is misleading. Because the separated lipid extracts were stained with ninhydrin, only nitrogen-containing lipids are visualized, and PC and CL would not be detected under these conditions. The same issue applies to figures labeled as showing “total lipids [%]”. These panels actually represent “[%] of amino lipids” and not total lipid composition, if I am not mistaken.

We appreciate and agree with the reviewer’s comment. We have revised the text (lines 125–127) to clarify that the lipids were detected under ninhydrin staining conditions and corrected the labels in Figs. 2E and 5C from “total lipids (%)” to “% of amino lipids”.

5. Page 6, line 189: A “not” is missing.

We have corrected the text in the revised manuscript.

6. Page 7, line 243: To the best of my knowledge, the membrane association of PmtA via amphipathic helices was first shown by Danne et al. (2015).

We thank the reviewer for pointing this out. We have added the reference of Danne et al. (2015) to the revised manuscript.

7. In the Introduction, the authors mention a paper on the N-terminal D1 domain structure of the Brucella abortus AcvB homolog VirJ (reference 20). It would be worthwhile to revisit this study in the Discussion and compare those findings with the result presented here. Brucella VirJ is not included in the sequence alignment in Fig. S4, and it is unclear where domain 1 of the Brucella protein is located compared to AcvB.

We thank the reviewer for this suggestion. In the revised manuscript, we have included *B. abortus* VirJ in the sequence alignment (now shown in Supplementary Fig. 8) and discussed this protein in the Discussion section (lines 298–299). As described in our response to Reviewer 1, Major comment #4, we also compared the N-terminal domain of AcvB with the D1 domain of *B. abortus* VirJ and found that they share structural similarity (r.m.s.d. 2.32 Å over 145 Cα atoms), indicating that the D1 domain corresponds to the N-terminal domain of AcvB. This comparison is now

described in the Results section (lines 89–91), and the structural superposition is shown in Supplementary Fig. 3.

Response to the Reviewer #3

Agrobacterium chromosomal virulence protein B (AcvB) is a periplasmic hydrolase involved in maintaining the level of lysyl-phosphatidylglycerol (Lys-PG) by catalyzing hydrolysis of Lys-PG into PG and lysine. The function of AcvB is crucial for the bacterial cells (A. tumefaciens) to adapt to acidic conditions and achieve optimal virulence. In the manuscript presented by Hoshi, M. and Watanabe Y., the crystal structures of mature AcvB (full length form lacking the signal sequence) and the C-terminal catalytic domain have been solved. The authors have identified a potential binding site for the Lys-PG head group in a negatively-charged cavity of the catalytic domain, analyzed the activities of a series of site-directed mutants in comparison with the wild type enzyme and detected a potential interaction between AcvB and LpiA. The results are new and beneficial for understanding the molecular mechanism underlying the catalytic function of AcvB.

The following are a few questions/comments for the authors to consider for further revision and improvement of their work.

We are grateful to the Reviewer 3 for the positive comments on our manuscript.

Major points:

1) In lines 85-87, the authors described the overall fold of the N-terminal domain of AcvB. What is the function of this domain? Does it also have the LysPG (or other substrate) hydrolyzing activity as the C-terminal domain, participate in substrate binding or regulate the catalytic activity of the C-terminal domain?

We thank the reviewer for this insightful comment. As described in our response to Reviewer 1, Major comment #2, we examined the Lys-PG hydrolase activities of the mature form of AcvB, its N-terminal domain, and its C-terminal domain (Supplementary Fig. 5 in the revised manuscript). While the C-terminal domain exhibited Lys-PG hydrolase activity, the N-terminal domain alone did not show detectable activity. Interestingly, the C-terminal domain displayed higher hydrolase activity than the mature form of AcvB, suggesting that the N-terminal domain may negatively regulate catalytic activity. In the crystal structure, the membrane-binding hydrophobic loop of the C-terminal domain interacts with the N-terminal domain of a neighboring AcvB molecule (Fig. 4A, B), raising the possibility that the N-terminal domain could occlude or restrain the membrane-interacting region. Such an arrangement would be consistent with the observed increase in activity upon N-terminal domain removal. Based on these observations, we speculate that the N-terminal domain could function as a regulatory module that modulates catalytic activity of the C-terminal domain. We have added this explanation to the revised manuscript (lines 261–267).

2) In lines 100-101, it was stated that the C-terminal domain of AcvB is necessary and sufficient for the Lys-PG hydrolase activity. How does the specific activity of the recombinant C-terminal domain of AcvB compare to the mature AcvB protein (with both N-terminal and C-terminal domains)? The comparison may also help to address the above question about whether the N-

terminal domain has a role in regulating the catalytic activity of the C-terminal domain.

Please see our above response.

3) In line 207-208, it was claimed that Trp378 and Leu379 contribute to the recognition of the acyl chains of Lys-PG through their hydrophobic side chains. However, I could not find any direct evidences supporting the conclusion. The data presented in Figure 5 only demonstrates that Trp378 and Leu379 are crucial for the activity of AcvB and association of the protein with membrane, but do not necessarily indicate that they are involved in recognizing the acyl chains of Lys-PG. To provide direct evidences, the authors could apply the protein lipid overlay assay (Dowler et al., Sci STKE 129:pl6, 2002) to analyze the interactions between AcvB (WT & mutant) and Lys-PG (and other lipids with different head group and acyl chain compositions).

We thank the reviewer for this important comment. Our original interpretation was based on the observation that the W378N/L379N substitution markedly reduced Lys-PG hydrolysis (Fig. 5C) while having minimal effects on hydrolysis of the headgroup analog LysEE (Fig. 5G, H), indicating that these residues are unlikely to be involved in headgroup recognition. Furthermore, the W378N/L379N substitution reduced membrane association (Fig. 5D, E), suggesting a role in membrane interaction. However, we agree that our data do not directly demonstrate specific recognition of the acyl chains of Lys-PG. Rather, the results support a role for these hydrophobic residues in membrane association. Although a protein lipid overlay assay could provide additional information on lipid binding, our data primarily indicate a role of these residues in membrane association rather than specific lipid recognition. We have therefore revised the manuscript to avoid overinterpretation and no longer describe these residues as directly recognizing the acyl chains of Lys-PG (Abstract in lines 9–10 and the Results section in lines 215–216).

4) The interaction between AcvB and LpiA discovered through the GST pull-down assays is very interesting. It will be great if the authors could measure the kinetic parameters of their interactions by using one of the quantitative methods, such as ITC, SPR or BLI.

We thank the reviewer for this helpful suggestion. Quantitative characterization of the interaction between AcvB and LpiA by methods such as ITC, SPR, or BLI would indeed be valuable. However, purified LpiA tends to aggregate at the high protein concentrations and near room temperature conditions typically required for these biophysical measurements. In addition, LpiA is an integral membrane protein and requires detergent for solubilization, which further complicates the preparation of stable samples suitable for quantitative interaction analyses. As a result, it was difficult to obtain stable samples appropriate for reliable ITC, SPR, or BLI measurements. Nevertheless, the reproducible GST pull-down assays consistently demonstrated the interaction between AcvB and LpiA, supporting the conclusion that these proteins associate with each other.

Minor points:

1) In Figure 3A, the distances (Å) between two hydrogen-bonded groups should be measured and labeled nearby the dash lines.

We thank the reviewer for this suggestion. We have labeled the distances between two hydrogen-bonded groups in Fig. 3A in the revised manuscript.

2) In the legends of Figures 2E, 3E, 5C, 5E, 5H, it was stated that the histograms are the ImageJ quantification results based on the data presented in Fig. 2D, 3D, 5B, 5D and 5G respectively. As there are three repeats for each data point, please clarify if the repeats are repeated measurements on the same TLC/western blot image or multiple repeats of the TLC/western blot experiments with one representative image shown in the figure.

We thank the reviewer for this comment. The quantification shown in the histograms was based on three independent TLC or western blot experiments. The images shown in the figures are representative images from these independent experiments. We have clarified this point in the figure legends and the Methods section in the revised manuscript (lines 471–472).

3) The gap between R_{work} and R_{free} for the structure of AcvB (25-456) is a bit too large (Table 1), maybe due to overfitting. The RMS deviation values of bond lengths and bond angles appear to be fairly low, as the stereochemistry might be too tightly restrained. The target weights could be optimized further to decrease the R_{free} or reduce the gap between R_{work} and R_{free} . It will be better to reduce the gap to around 3% if possible.

We thank the reviewer for this helpful suggestion. Following the reviewer's advice, we re-refined the structure with optimized target weights. This resulted in slightly increased R_{work} and R_{free} values but reduced the gap between them (from 0.212/0.277 to 0.225/0.280). The updated refinement statistics and Ramachandran analysis are reported in Table 1 and the Methods section in the revised manuscript (lines 387–388).

4) There are several typos in the manuscript to be fixed, such as "decreasingin their membrane" (line 33-34) and "whereas the W378F/L379F mutant did (not)" (line 189). Please check the manuscript carefully to minimize typo errors.

We thank the reviewer for pointing out these errors. We have corrected the errors mentioned by the reviewer and carefully checked the entire manuscript to minimize additional errors.